# Federated Frank-Wolfe Algorithm

**Ali Dadras**
Umeå University
ali.dadras@umu.se

**Karthik Prakhya**
Umeå University
karthik.prakhya@umu.se

**Alp Yurtsever**
Umeå University
alp.yurtsever@umu.se

## Abstract

Federated learning (FL) has gained much attention in recent years for building privacy-preserving collaborative learning systems. However, FL algorithms for constrained machine learning problems are still very limited, particularly when the projection step is costly. To this end, we propose a Federated Frank-Wolfe Algorithm (FEDFW). FEDFW provably finds an $\varepsilon$-suboptimal solution of the constrained empirical risk-minimization problem after $\mathcal{O}(\varepsilon^{-2})$ iterations if the objective function is convex. The rate becomes $\mathcal{O}(\varepsilon^{-3})$ if the objective is non-convex. The method enjoys data privacy, low per-iteration cost and communication of sparse signals. We demonstrate empirical performance of the FEDFW algorithm on several machine learning tasks.

## 1 Introduction

This paper introduces a novel variant of the Frank-Wolfe Algorithm, FEDFW, designed for the increasingly popular Federated Learning (FL) paradigm in machine learning. We focus on the constrained Empirical Risk Minimization (ERM) template with $n$ clients:

$$\min_{\mathbf{x} \in \mathcal{D}} \quad F(\mathbf{x}) := \frac{1}{n} \sum_{i=1}^{n} f_i(\mathbf{x}), \tag{1}$$

where domain $\mathcal{D} \subseteq \mathbb{R}^p$ is a convex and compact set (with diameter $D := \max_{\mathbf{x}, \mathbf{y} \in \mathcal{D}} \|\mathbf{x} - \mathbf{y}\|$), and the clients' loss functions $f_i : \mathbb{R}^p \to \mathbb{R}$ (for $i = 1, \ldots, n$) are $L$-smooth (*i.e.,* their gradients are Lipschitz continuous with constant $L$).

FL holds great promise for solving optimization problems over large networks, where clients collaborate under the orchestration of a server for finding a good global model. FL methods iteratively alternate between training and aggregation steps. Training steps are performed locally by the clients. Each participating client updates their local model by taking one or several training steps with their local data. Then, in the aggregation step, participating clients transfer their local models (but not their data) to the server. The server maintains and updates the global model and broadcasts it to the clients.

The vast majority of FL algorithms focus on unconstrained optimization problems. Their extensions for constrained problems require projection at each local training step. In many machine learning applications, however, the cost of projection causes a computational bottleneck that prevents us from solving these problems at a realistic scale. Frank-Wolfe Algorithm (FW) [Frank and Wolfe, 1956] has been the method of choice in the machine learning community for solving such problems in the centralized setting. The main workhorse of FW is the so-called linear minimization oracle,

$$\text{lmo}(\mathbf{y}) = \arg\min_{\mathbf{x} \in \mathcal{D}} \langle \mathbf{y}, \mathbf{x} \rangle, \tag{2}$$

which is often cheaper than the projection. A popular example is the nuclear-norm constraint. The projection onto a nuclear-norm ball requires a full-spectrum singular value decomposition. In contrast,

Workshop on Federated Learning: Recent Advances and New Challenges, in Conjunction with NeurIPS 2022 (FL-NeurIPS'22). This workshop does not have official proceedings and this paper is non-archival.

linear minimization amounts to finding the top singular vector, which we can approximate efficiently by using power method.

Intriguingly, FW is not yet studied for FL. This paper takes the initial steps to bridge this gap.

**Contributions and roadmap.** With the above motivation, we present FEDFW, a novel FW variant for FL. The paper is organized as follows: Section 2 presents the review of the literatures on FL and FW. It is surprising that the literature lacks a FW variant for FL. We try to explain this in Section 3 by presenting an example where an obvious federated FW extension (*i.e.,* standard FW plus aggregation step) fails. We introduce FEDFW in Section 4. FEDFW does not replace clients' local models by the global model sent by the server. Instead, it penalizes clients' loss functions for the distance between the global model and their local models. We present convergence guarantees of FEDFW in Section 4.1. The method provably finds an $\varepsilon-$suboptimal solution after $\mathcal{O}(\varepsilon^{-2})$ iterations if the objective function is smooth and convex, see Theorem 1. When the objective is non-convex, the complexity becomes $\mathcal{O}(\varepsilon^{-3})$, see Theorem 2. Section 5 presents preliminary numerical experiments on various machine learning tasks with convex and non-convex objective functions. Finally, Section 6 draws conclusions with some discussion on the limitations of the proposed method. Proofs and technical details are deferred to the appendices.

## 2   Related Work

**Federated Learning.** FEDAVG has been the main corner stone for the recent FL literature because of its practical capability of handling different concerns and issues such as *privacy and security*, *data heterogeneity*, *computational costs*. It is introduced to train deep networks based on iterative model averaging [McMahan et al., 2017, Konečnỳ et al., 2016]. Although, it is evident that fix points of some variants of FEDAVG need not converge to the minimizer of $F(\mathbf{x})$ even in the least squares problem [Pathak and Wainwright, 2020], and even diverging Zhang et al. [2020], performance of FEDAVG has been studied widely under different assumptions, see [Yu et al., 2019, Woodworth et al., 2020a] for homogeneous i.i.d. data assumption, and Woodworth et al. [2020b], Li et al. [2019], Sahu et al. [2018] for non-independent data and various heterogeneity assumptions, see also Stich [2018] for a convergence analysis of FEDAVG. Haddadpour et al. [2019] provides convergence analysis of FEDAVG for non-convex objectives satisfying Polyak-Lojasiewicz (PL) condition condition. [Al-Shedivat et al., 2020] analyzed FEDAVG from probabilistic inference perspective. Generalizing FEDAVG, one can consider different update rules for client or server.

**Frank-Wolfe Algorithm.** FW (aka, conditional gradient method or CGM) is introduced in [Frank and Wolfe, 1956] for minimization of a convex quadratic objective over a polytope constraint. Its analysis is later extended for general convex objectives and arbitrary convex and compact sets in [Levitin and Polyak, 1966]. The method became popular in machine learning applications following the seminal works of [Hazan and Kale, 2012a] and [Jaggi, 2013].

The ever-increasing interest in FW for data science applications motivated development of new results and new variants. Faster rates are shown for FW when $\mathcal{D}$ is a strongly convex set [Garber and Hazan, 2015]. [Lacoste-Julien, 2016] proved that FW finds a stationary point when the objective function is non-convex. Online, stochastic, and variance reduced stochastic variants of FW are proposed, starting with [Hazan and Kale, 2012b] and [Hazan and Luo, 2016] for convex objectives and with [Reddi et al., 2016] for non-convex. FW is combined with Nesterov smoothing [Nesterov, 2005] for non-smooth and composite objectives, respectively in [Lan, 2012] and [Yurtsever et al., 2018]. FW for problems with affine equality constraints are introduced in [Gidel et al., 2018] and [Yurtsever et al., 2019] based on augmented Lagrangian penalty.

There are also design variants of FW for making better use of computational resources in certain cases. Examples include (but are not limited to) the well-known away-step [Guélat and Marcotte, 1986] and pairwise step FW [Lacoste-Julien and Jaggi, 2015], FW with in-face directions [Freund et al., 2017], FW with lazy [Braun et al., 2017] or blended updates [Braun et al., 2019], FW with line-search [Pedregosa et al., 2020], FW with a restarting scheme [Kerdreux et al., 2019], FW with sketching for better storage costs [Yurtsever et al., 2017], and the conditional gradient sliding which reduces the number of gradient evaluations by reusing past gradients [Lan and Zhou, 2016].

The most closely related methods to our work are the distributed FW variants. However, the variants in [Wai et al., 2017], [Mokhtari et al., 2018], [Gao et al., 2021] are fundamentally different than FEDFW

as they require sharing gradient information of the clients with the server or with the neighboring nodes. In FEDFW, clients do not share gradients, which is critical for data privacy [Li et al., 2022].

Other distributed FW variants are proposed in [Zheng et al., 2018], [Wang et al., 2016], and [Zhang et al., 2021]. However, the method proposed by Zheng et al. [2018] is limited to the covex low-rank matrix optimization problem, and the methods in [Wang et al., 2016] and Zhang et al. [2021] assume that the problem domain is block spearable.

## 3 Preliminaries

The obvious thing to try for extending FW for federated learning is combining local FW steps with an aggregation step. That is, for $t = 1, \ldots, T$, perform the following procedure:

$$
\begin{aligned}
&[\text{local training}] && \mathbf{s}_i^t \in \arg\min_{\mathbf{x} \in \mathcal{D}} \; \langle \nabla f_i(\bar{\mathbf{x}}^t), \mathbf{x} \rangle && \text{for } i = 1, \ldots, n \\
&[\text{local training}] && \mathbf{x}_i^{t+1} = (1 - \eta_t)\bar{\mathbf{x}}^t + \eta_t \mathbf{s}_i^t && \text{for } i = 1, \ldots, n \\
&[\text{aggregation}] && \bar{\mathbf{x}}^{t+1} = \frac{1}{n} \sum_{i=1}^{n} \mathbf{x}_i^{t+1}
\end{aligned}
\tag{3}
$$

However, this algorithm may fail to find a solution of (1) even for a simple 1-dimensional example. We demonstrate this in an example in Appendix A.

Why does the above method fail? Consider the following reformulation of problem (1):

$$
\min_{\mathbf{x}_i \in \mathcal{D}} \frac{1}{n} \sum_{i=1}^{n} f_i(\mathbf{x}_i) \quad \text{subj.to} \quad \mathbf{x}_1 = \mathbf{x}_2 = \ldots = \mathbf{x}_n.
\tag{4}
$$

This is an equivalent formulation but written in terms of the clients' local variables with a consensus constraint $\mathcal{C} := \{[\mathbf{x}_1, \ldots, \mathbf{x}_n] \in \mathbb{R}^{d \times n} : \mathbf{x}_1 = \mathbf{x}_2 = \ldots = \mathbf{x}_n\}$. Many algorithms in FL can be viewed as special instances of classic optimization methods applied to (4) (for example, FEDAVG is the projected gradient method applied to (4)). From this perspective, model aggregation is just a projection onto $\mathcal{C}$. This approach fails for FW, because FW is a projection-free algorithm, hence the standard aggregation techniques are not suitable. Nevertheless, formulation (4) is still useful and we design a functional FW method for FL based on this formulation in the next section.

## 4 Federated Frank-Wolfe Algorithm

Even though FW is not qualified for an immediate extension with the standard aggregation step, we can still design a federated FW algorithm by applying an appropriate FW variant to the problem (4). The literature contains several FW variants which can handle linear equality constraints efficiently, including the ones in [Gidel et al., 2018], [Yurtsever et al., 2018], [Liu et al., 2019] and [Yurtsever et al., 2019]. Motivated by its simplicity and the strong performance profile, we focus on the approach presented in [Yurtsever et al., 2018]. This method is proposed for minimizing a generic composite convex objective function $f + g$, where $f$ and $g$ denote the smooth and non-smooth components. We review it here for our problem with the consensus constraint, which falls into this composite template in terms of the concatenated variable $\mathbf{X} := [\mathbf{x}_1, \mathbf{x}_2, \ldots, \mathbf{x}_n]$ as

$$
\min_{\mathbf{X} \in \mathcal{D}^n} \frac{1}{n} \sum_{i=1}^{n} f_i(\mathbf{X}\mathbf{e}_i) + \delta_{\mathcal{C}}(\mathbf{X}),
\tag{5}
$$

where $\mathbf{e}_i$ denotes the $i$th standard unit vector, and $\delta_{\mathcal{C}}$ is the indicator function for the consensus.

The original FW is not applicable since the objective function is non-smooth due to the indicator function. The main idea is to perform FW updates on a surrogate objective which replaces the hard constraint $\delta_{\mathcal{C}}$ with a smooth function that penalizes squared distance between $\mathbf{X}$ and the consensus set $\mathcal{C}$. To this end, at iteration $t$, we take a FW step with respect to the following surrogate function:

$$
\hat{F}_t(\mathbf{X}) = \frac{1}{n} \sum_{i=1}^{n} f_i(\mathbf{X}\mathbf{e}_i) + \frac{\lambda_t}{2} \text{dist}(\mathbf{X}, \mathcal{C})^2, \quad \text{where } \lambda_t > 0 \text{ is the penalty parameter.}
\tag{6}
$$

Then, the linear minimization oracle is

$$\mathbf{S}^t \in \arg\min_{\mathbf{X} \in \mathcal{D}^n} \ \langle \nabla \hat{F}_t(\mathbf{X}^t), \mathbf{X} \rangle. \tag{7}$$

Since $\mathcal{D}^n$ is separable for the columns of $\mathbf{X}$, we can evaluate (7) in parallel for $\mathbf{x}_1, \mathbf{x}_2, \ldots, \mathbf{x}_n$:

$$\mathbf{S}^t = \sum_{i=1}^n \mathbf{s}_i^t \cdot \mathbf{e}_i^\top, \quad \text{where} \quad \mathbf{s}_i^t \in \arg\min_{\mathbf{x} \in \mathcal{D}} \ \langle \frac{1}{n} \nabla f_i(\mathbf{x}_i^t) + \lambda_t(\mathbf{x}_i^t - \bar{\mathbf{x}}^t), \mathbf{x} \rangle, \ \bar{\mathbf{x}}^t := \frac{1}{n} \sum_{i=1}^n \mathbf{x}_i^t. \tag{8}$$

Finally, we update our estimation as $\mathbf{X}^{t+1} = (1 - \eta_t)\mathbf{X}^t + \eta_t \mathbf{S}^t$, columns of which can be computed in parallel by

$$\mathbf{x}_i^{t+1} = (1 - \eta_t)\mathbf{x}_i^t + \eta_t \mathbf{s}_i^t \quad \text{for some step-size } \eta_t \in [0, 1]. \tag{9}$$

Notice that communication is needed only for computing $\bar{\mathbf{x}}$, which corresponds to an aggregation step, and everything else can be computed locally by the clients. We call this method as Federated Frank-Wolfe Algorithm (FEDFW).

---

**Algorithm 1** FEDFW: Federated Frank-Wolfe Algorithm

---

**set** $\mathbf{x}_i^1 \in \mathbb{R}^p$, $\forall i \in [n]$, $\lambda_t$, $\eta_t$, $\bar{\mathbf{x}}^1 = \frac{1}{n} \sum_{i=1}^n \mathbf{x}_i^1$
**for** round $t = 1, 2, \ldots, T$ **do**

    — **Client**-level local training ————————————
    **for** client $i = 1, 2, \ldots, n$ **do**
        $\mathbf{g}_i^t = \frac{1}{n} \nabla f_i(\mathbf{x}_i^t) + \lambda_t(\mathbf{x}_i^t - \bar{\mathbf{x}}^t)$
        $\mathbf{s}_i^t = \arg\min\{\langle \mathbf{g}_i^t, \mathbf{x} \rangle : \mathbf{x} \in \mathcal{D}\}$
        $\mathbf{x}_i^{t+1} = (1 - \eta_t)\mathbf{x}_i^t + \eta_t \mathbf{s}_i^t$
        Client communicates $\mathbf{s}_i^t$ to the server.
    **end for**

    — **Server**-level aggregation ————————————
    $\bar{\mathbf{x}}^{t+1} = (1 - \eta_t)\bar{\mathbf{x}}^t + \eta_t \left( \frac{1}{n} \sum_{i=1}^n \mathbf{s}_i^t \right)$
    Server communicates $\bar{\mathbf{x}}^{t+1}$ to the clients.

**end for**

---

### 4.1 Convergence Guarantees

This section presents convergence guarantees of FEDFW.

**Theorem 1** (Convex setup). *Consider problem* (1) *with $L$-smooth and convex loss functions $f_i$. Then, estimation $\bar{\mathbf{x}}^t$ generated by FEDFW with step-size $\eta_t = \frac{2}{t+1}$, and penalty parameter $\lambda_t = \lambda_0 \sqrt{t+1}$ for some $\lambda_0 > 0$ satisfies*

$$F(\bar{\mathbf{x}}^t) - F^* \leq \frac{2D^2 L}{t} + \frac{1}{\sqrt{t}} \left( 2nD^2\lambda_0 + \frac{2G}{\sqrt{n}\lambda_0} \left( \|\mathbf{Y}^\star\| + D\sqrt{\lambda_0(L + n\lambda_0)} \right) \right), \tag{10}$$

*where $\mathbf{Y}^\star$ denotes an arbitrary solution to the dual of Problem* (4).

The proof of Theorem 1 largely follows from Theorem 3 in [Yurtsever et al., 2018]. For the sake of completeness, we present the details in Appendix B.

Next, we present convergence guarantees of FEDFW for non-convex problems.

**Theorem 2** (Non-convex setup). *Consider problem* (1) *with $L$-smooth and non-convex loss functions $f_i$. Suppose that the sequence $\{\bar{\mathbf{x}}^t\}$ is generated by FEDFW with the fixed step-size $\eta_t = T^{-2/3}$, and penalty parameter $\lambda_t = \lambda_0 T^{1/3}$ for some $\lambda_0 > 0$. Then,*

$$\min_{1 \leq t \leq T} \ \max_{\mathbf{u} \in \mathcal{D} \cap \mathcal{C}} \left\{ \langle \nabla F(\bar{\mathbf{x}}^t), \bar{\mathbf{x}}^t - \mathbf{u} \rangle \right\} \leq \mathcal{O}(T^{-1/3}). \tag{11}$$

We present the proof of Theorem 2 in Appendix C. Note that our analysis in this setting is entirely novel and nonconvex objectives are not studied in [Yurtsever et al., 2018].

## 4.2 FEDFW with augmented Lagrangian

Yurtsever et al. [2019] presents an extension of [Yurtsever et al., 2018] with an augmented Lagrangian dual step. While the original method offers tighter bounds in theory, the new variant often performs better in practice. Motivated by this extension, we present a practical variant of the federated Frank-Wolfe algorithm with augmented Lagrangian dual steps, FEDFW+, in Appendix D. We compare the empirical performance of FEDFW and FEDFW+ in the next section.

## 5 Numerical Experiments

This section validates our theory and demonstrates the empirical performance of the FEDFW algorithm on various convex and non-convex optimization problems. All the experiments were run on a laptop with an Intel i9-10855H processor with 2.40GHz clock speed and with 32 GB of RAM.

### 5.1 LASSO for Sparse Feature Recovery

First, we consider $\ell_1$-regularized least squares formulation with a synthetic dataset:

$$\min_{\mathbf{x}} \quad \|\mathbf{A}\mathbf{x} - \mathbf{b}\|^2 \quad \text{subj.to} \quad \|\mathbf{x}\|_1 \leq \alpha, \tag{12}$$

where $\mathbf{A}$ is a random $200 \times 400$ matrix with entries drawn from a Gaussian distribution with a diagonal covariance matrix whose diagonal entries are sampled uniformly random from $[1, 2]$ interval. The ground truth weight vector $\mathbf{x}^*$ is an s-sparse random vector with non-zero entries drawn from the standard normal distribution, and $\mathbf{b} = \mathbf{A}\mathbf{x}^*$. We run FEDFW and FEDFW+ Algorithms for $10^5$ iterations with $\lambda_0^{\text{FedFW}} = 5, \lambda_0^{\text{FedFW+}} = 0.5$. Figure 1a shows how objective residual $F(\bar{\mathbf{x}}^t) - F^*$ changes with respect to the iteration counter and cpu time for FEDFW and FEDFW+ algorithms.

### 5.2 Sparse Multi-Class Logistic Regression for Classification

Next, we consider $\ell_1$-constrained multi-class logistic regression model for UCI ML hand-written digits dataset [Kaynak, 1995]. The goal is to train a model to recognize hand-written digits from 0 to 9 (see [Kaynak, 1995] for more information on the dataset). Similar to the previous experiment with LASSO, we ran both the FEDFW and FEDFW+ algorithm for $10^5$ iterations with $\lambda_0^{\text{FedFW}} = \lambda_0^{\text{FedFW+}} = 4 \times 10^{-4}$. Figure 1b shows the logarithmic scale plot of the objective residual versus the number of iterations and cpu time.

### 5.3 Matrix Completion with MovieLens Dataset

For this subsection, we discuss how the FEDFW algorithm can be applied to the problem of matrix completion with the MovieLens100k dataset [Harper and Konstan, 2015]. The goal of the matrix completion problem is to estimate the elements of a matrix given the some of the entries.

For $r$ customers and $q$ products and a partially completed preference matrix $Y \in \mathbb{R}^{r \times q}$, the matrix completion problem can be stated as follows:

$$\min_{\mathbf{X} \in \mathbb{R}^{r \times q}} \sum_{i=1}^{r} \sum_{j=1}^{q} (X_{ij} - Y_{ij})^2 \quad \text{subj.to} \quad \|\mathbf{X}\|_* \leq \alpha,$$

where $\|\cdot\|_*$ denotes the nuclear norm (sum of singular values) and $\alpha > 0$ is a model parameter.

We ran both FEDFW and FEDFW+ Algorithms for $2 \times 10^3$ iterations with $\lambda_0^{\text{FedFW}} = \lambda_0^{\text{FedFW+}} = 10^{-5}$ and $\alpha = 7000$ and 40 clients. The ratings were shuffled and evenly distributed between the clients. Figure 1c shows the evaluation root mean squared error (RMSE) on the train and test partitions as a function of the iteration counter and cpu time.

### 5.4 Quadratic Assignment Programming (QAP) with Birkhoff Constraint

Finally, we test our methods on a relaxation of the quadratic assignment programming (QAP) proposed in [Vogelstein et al., 2015]. The formulation is as follows:

$$\min_{\mathbf{X}} \quad \text{trace}(\mathbf{A}\mathbf{X}\mathbf{B}^T\mathbf{X}^T) \quad \text{subj.to} \quad \mathbf{X} \in [0, 1]^{q \times q}, \mathbf{X}\mathbf{1} = \mathbf{X}^T\mathbf{1} = \mathbf{1}, \tag{13}$$

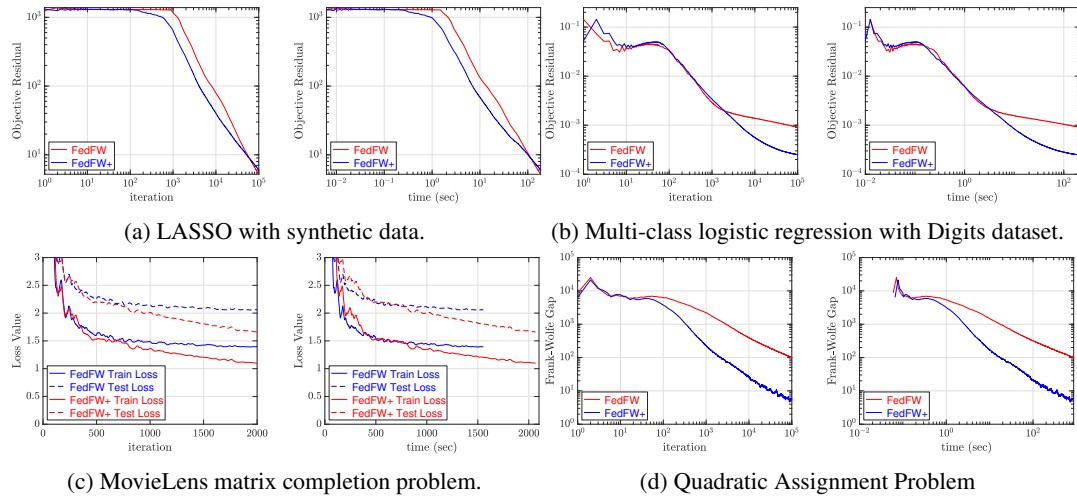

(a) LASSO with synthetic data.

(b) Multi-class logistic regression with Digits dataset.

(c) MovieLens matrix completion problem.

(d) Quadratic Assignment Problem

Figure (1)    Empirical comparison of FEDFW and FEDFW+ in various federated learning problems.

here, $\mathbf{A}$ and $\mathbf{B}$ are given cost matrices (called as the flow and distance matrix in QAP) and $\mathbf{1}$ denotes $q$-dimensional vector of ones. This is a non-convex optimization problem over the convex hull of permutation matrices, also known as the Birkhoff polytope. We use CHR12A dataset from QAPLIB [Burkard et al., 1997]. We use both FEDFW and FEDFW+ algorithms with 72 clients and $10^5$ iterations. We set $\lambda_0^{\text{FedFW}} = 65$ and $\lambda_0^{\text{FedFW+}} = 10$.

## 6    Conclusions

We introduced FEDFW Algorithm for the constrained minimization of a convex or non-convex function and established convergence guarantees for this algorithm. FEDFW guarantees $\mathcal{O}(t^{-1/2})$ and $\mathcal{O}(t^{-1/3})$ convergence rates for the convex and non-convex objective functions, respectively. We also proposed an empirically faster version of FEDFW with augmented Lagrangian dual updates.

We conclude with a short discussion on the opportunities and limitations of our study. In FEDFW, clients communicate the output of their linear minimization oracle, which is a nonlinear operator and its reverse operator is ill-conditioned. In general, it is not possible to recover the gradient from its linear minimization output even if $\mathcal{D}$ is known. For example, if $\mathcal{D}$ is the $\ell_1$ norm-ball, then $\mathbf{s}_i^t$ reveals only the sign of the maximum entry of the gradient. Or, if $\mathcal{D}$ is the nuclear norm-ball, then $\mathbf{s}_i^t$ reveals only the top eigenvectors of the gradient. The nonlinearity of the FW oracle may improve the privacy in federated learning. Moreover, FEDFW offers low communication overhead since the communicated signals are the extreme points of $\mathcal{D}$, which typically have a low dimensional representation. We plan to analyze communication cost and privacy of FEDFW in a future-work.

This work is still in its early stage. In this paper, we focused on a basic variant of FEDFW which performs only one local step at each communication round. Moreover, we did not study the stochastic variant of FEDFW. In a realistic FL framework, however, only a subset of clients can participate in each communication round. Also, each client can perform different number of iterations in each round. We plan to extend FEDFW in these directions in the near future. We also plan to perform extensive numerical experiments with comparisons against existing methods in the literature. We are unaware of any existing FW variant for FL but we will include performance comparisons against distributed and decentralized FW variants and a number of baseline heuristics (including FW with aggregation step and FEDAVG with projection step) and some operator splitting methods for federated/distributed learning [Tran Dinh et al., 2021].

## Acknowledgments

This project is supported by the Wallenberg AI, Autonomous Systems and Software Program (WASP) funded by the Knut and Alice Wallenberg Foundation. The computations were enabled by resources provided by the Swedish National Infrastructure for Computing (SNIC) at Chalmers Centre for Computational Science and Engineering (C3SE) partially funded by the Swedish Research Council through grant agreement no. 2018-05973.

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

# A    FW with simple aggregation

The trivial extension of FW with an aggregation step (see (3)) fails for the following 1-dimensional example:

**Example 1.** Consider the following instance of model problem (1):

$$\min_{x \in [-1,1]} \quad \frac{1}{2}(x-3)^2 + \frac{1}{2}(x+1)^2. \tag{14}$$

Here, $f_1(x) = (x-3)^2$ and $f_2(x) = (x+1)^2$. It is easy to check that the unique solution of this problem is $x^\star = 1$.

Now, consider procedure (3) initialized from $\bar{x}^1 = 0$. Then, $s_1^1 = 1$ and $s_2^1 = -1$, which gives

$$\bar{x}^2 = \frac{1}{2}x_1^2 + \frac{1}{2}x_2^2 = (1 - \eta_1)\bar{x}^1 + \eta_1 \frac{1}{2}(s_1^1 + s_2^1) = 0, \qquad \forall \eta_1 \in [0, 1]. \tag{15}$$

Therefore, $x = 0$ is a fixed point for the procedure (3) although the unique solution of (14) is $x = 1$. We conclude that this method may fail. Note that similar arguments hold even if we run multiple local training steps before the aggregation.

# B    Convergence Analysis of Algorithm 1 (convex case)

In this section we provide the proof of Theorem 1. To start with, we mention some remarks.

**Remark 1.** *Gradient of the surrogate function* (6) *is given by*

$$
\begin{aligned}
\nabla \hat{F}_t(\mathbf{X}) &= \frac{1}{n} \sum_{i=1}^{n} \nabla f_i(\mathbf{X}\mathbf{e}_i) \cdot \mathbf{e}_i^\top + \lambda_t \left( \mathbf{X} - \text{proj}_\mathcal{C}(\mathbf{X}) \right) \\
&= \frac{1}{n} \sum_{i=1}^{n} \nabla f_i(\mathbf{x}_i) \cdot \mathbf{e}_i^\top + \lambda_t \sum_{i=1}^{n} (\mathbf{x}_i - \bar{\mathbf{x}}) \cdot \mathbf{e}_i^\top \quad \text{with} \quad \bar{\mathbf{x}} := \frac{1}{n} \sum_{i=1}^{n} \mathbf{x}_i.
\end{aligned}
\tag{16}
$$

**Remark 2.** *Assuming $f_i(.)$ are L-smooth, $F(.)$ is L-smooth.*

**Remark 3.** *Assuming $f_i(.)$ are convex, $F(.)$ and $\hat{F}(.)$ is also convex.*

**Remark 4.** *One can simply prove $\frac{\beta}{2} \leq \frac{\gamma}{2} + \frac{\beta}{2}\left(\frac{\beta}{\gamma} - 1\right)$.*

**Remark 5.** *$\hat{F}_t(\mathbf{X})$ is $\hat{L}_t := \left(\frac{L}{n} + \lambda_t\right)$- smooth.*

*Proof.* Using Equation (16), we have

$$\|\nabla\hat{F}_t(\mathbf{X}) - \nabla\hat{F}_t(\mathbf{Y})\|_F = \|\frac{1}{n}\sum_{i=1}^{n}(\nabla f_i(\mathbf{x}_i) - \nabla f_i(\mathbf{y}_i))\cdot\mathbf{e}_i^\top + \lambda_t\sum_{i=1}^{n}(\mathbf{x}_i - \bar{\mathbf{x}} - \mathbf{y}_i + \bar{\mathbf{y}})\cdot\mathbf{e}_i^\top\|_F$$

$$\leq \frac{1}{n}\|\sum_{i=1}^{n}(\nabla f_i(\mathbf{x}_i) - \nabla f_i(\mathbf{y}_i))\cdot\mathbf{e}_i^\top\|_F + \lambda_t\|\sum_{i=1}^{n}(\mathbf{x}_i - \bar{\mathbf{x}} - \mathbf{y}_i + \bar{\mathbf{y}})\cdot\mathbf{e}_i^\top\|_F$$

$$\leq \frac{1}{n}\sqrt{\|\sum_{i=1}^{n}(\nabla f_i(\mathbf{x}_i) - \nabla f_i(\mathbf{y}_i))\cdot\mathbf{e}_i^\top\|_F^2} + \lambda_t\|\sum_{i=1}^{n}(\mathbf{x}_i - \bar{\mathbf{x}} - \mathbf{y}_i + \bar{\mathbf{y}})\|_2$$

$$\leq \frac{1}{n}\sqrt{\sum_{i=1}^{n}\|\nabla f_i(\mathbf{x}_i) - \nabla f_i(\mathbf{y}_i)\|_F^2} + \lambda_t\|\sum_{i=1}^{n}(\mathbf{x}_i - \bar{\mathbf{x}} - \mathbf{y}_i + \bar{\mathbf{y}})\|_2$$

$$\leq \frac{L}{n}\sqrt{\sum_{i=1}^{n}\|\mathbf{x}_i - \mathbf{y}_i\|_F^2} + \lambda_t\|\sum_{i=1}^{n}(\mathbf{x}_i - \bar{\mathbf{x}} - \mathbf{y}_i + \bar{\mathbf{y}})\|_2$$

$$= \frac{L}{n}\|\mathbf{X} - \mathbf{Y}\|_F + \lambda_t\|\mathbf{X} - \bar{\mathbf{X}} - \mathbf{Y} + \bar{\mathbf{Y}}\|_F$$

$$\leq \frac{L}{n}\|\mathbf{X} - \mathbf{Y}\|_F + \lambda_t\|(\mathbf{X} - \mathbf{Y})(I_n - \frac{1}{n}J_n)\|_F$$

$$\leq \frac{L}{n}\|\mathbf{X} - \mathbf{Y}\|_F + \lambda_t\|(\mathbf{X} - \mathbf{Y})\|_F$$

$$= (\frac{L}{n} + \lambda_t)\|\mathbf{X} - \mathbf{Y}\|_F. \tag{17}$$

$\square$

where $J_n$ is all-ones matrix of size $n \times n$ and in the $8^{th}$ line, we used the fact that the maximum eigenvalue of the centering matrix $K = I_n - \frac{1}{n}J_n$ is 1.

## B.1 Proof of Theorem 1

Algorithm 1 is a special case of Algorithm 2 in [Yurtsever et al., 2018] Therefore, using Theorem 3.3 in [Yurtsever et al., 2018] for smooth and convex $F$, we have

$$F(\bar{\mathbf{x}}^t) - F^* \leq \frac{2D^2 L}{t} + \frac{1}{\sqrt{t}}\left(2nD^2\lambda_0 + \frac{2G}{\sqrt{n}\lambda_0}\left(\|\mathbf{Y}^\star\| + D\sqrt{\lambda_0(L + n\lambda_0)}\right)\right). \tag{18}$$

For the sake of completenes we also bring the proof.

$$\hat{F}_t(\mathbf{X}^{t+1}) \leq \hat{F}_t(\mathbf{X}^t) + \langle\nabla\hat{F}_t(\mathbf{X}^t), \mathbf{X}^{t+1} - \mathbf{X}^t\rangle + \frac{\hat{L}_t}{2}\|\mathbf{X}^{t+1} - \mathbf{X}^t\|_F^2$$

$$= \hat{F}_t(\mathbf{X}^t) + \eta_t\langle\nabla\hat{F}_t(\mathbf{X}^t), \mathbf{S}^t - \mathbf{X}^t\rangle + \frac{\hat{L}_t\eta_t^2}{2}\|\mathbf{S}^t - \mathbf{X}^t\|_F^2$$

$$\leq \hat{F}_t(\mathbf{X}^t) + \eta_t\langle\nabla\hat{F}_t(\mathbf{X}^t), \mathbf{S}^t - \mathbf{X}^t\rangle + \frac{\hat{L}_t\eta_t^2}{2}nD^2 \tag{19}$$

Now, let us define $\hat{F}(\mathbf{X}) := \frac{1}{n}\sum_{i=1}^{n}f_i(\mathbf{X}\mathbf{e}_i)$. By definition of $\mathbf{S}^t$ and convexity of $\hat{F}(.)$, we get

$$\langle\nabla\hat{F}_t(\mathbf{X}^t), \mathbf{S}^t - \mathbf{X}^t\rangle \leq \langle\nabla\hat{F}_t(\mathbf{X}^t), \mathbf{X}^* - \mathbf{X}^t\rangle$$

$$= \langle\nabla\hat{F}(\mathbf{X}^t), \mathbf{X}^* - \mathbf{X}^t\rangle + \langle\nabla(\frac{\lambda_t}{2}\text{dist}(\mathbf{X}^t, \mathcal{C})^2), \mathbf{X}^* - \mathbf{X}^t\rangle$$

$$\leq \hat{F}(\mathbf{X}^*) - \hat{F}(\mathbf{X}^t) + \left(\underbrace{\delta_{\mathcal{C}}(\mathbf{X}^*)}_{=0\ (X^*\in\mathcal{C})} - \lambda_t\,\text{dist}(\mathbf{X}^t, \mathcal{C})^2\right)$$

$$= \hat{F}(\mathbf{X}^*) - \hat{F}_t(\mathbf{X}^t) - \frac{\lambda_t}{2}\text{dist}(\mathbf{X}^t, \mathcal{C})^2 \tag{20}$$

where in the third line we used Lemma 10 (d) from [Tran-Dinh et al., 2018]. Combining Equation (19), and Equation (20), we get

$$\hat{F}_t(\mathbf{X}^{t+1}) \leq \hat{F}_t(\mathbf{X}^t) + \eta_t \left( \hat{F}(\mathbf{X}^*) - \hat{F}_t(\mathbf{X}^t) - \frac{\lambda_t}{2} \mathrm{dist}(\mathbf{X}^t, \mathcal{C})^2 \right) + \frac{\hat{L}_t \eta_t^2}{2} n D^2, \quad (21)$$

Using Remark 4 with $\beta = \lambda_t, \gamma = \lambda_{t-1}$, we have

$$\hat{F}_t(\mathbf{X}^t) = \hat{F}(\mathbf{X}^t) + \frac{\lambda_t}{2} \mathrm{dist}(\mathbf{X}^t, \mathcal{C})^2$$

$$\leq \hat{F}(\mathbf{X}^t) + \frac{\lambda_{t-1}}{2} \mathrm{dist}(\mathbf{X}^t, \mathcal{C})^2 + \frac{\lambda_t}{2} \left( \frac{\lambda_t}{\lambda_{t-1}} - 1 \right) \mathrm{dist}(\mathbf{X}^t, \mathcal{C})^2$$

$$= \hat{F}_{t-1}(\mathbf{X}^t) + \frac{\lambda_t}{2} \left( \frac{\lambda_t}{\lambda_{t-1}} - 1 \right) \mathrm{dist}(\mathbf{X}^t, \mathcal{C})^2. \quad (22)$$

Substituting Equation (22) into Equation (21) and subtracting $\hat{F}(\mathbf{X}^*)$ from both sides, we have

$$\hat{F}_t(\mathbf{X}^{t+1}) - \hat{F}(\mathbf{X}^*) \leq (1 - \eta_t) \left( \hat{F}_t(\mathbf{X}^t) - \hat{F}(\mathbf{X}^*) \right) - \frac{\eta_t \lambda_t}{2} \mathrm{dist}(\mathbf{X}^t, \mathcal{C})^2 + \frac{\hat{L}_t \eta_t^2}{2} n D^2$$

$$= (1 - \eta_t) \left( \hat{F}_{t-1}(\mathbf{X}^t) - \hat{F}(\mathbf{X}^*) \right) + \frac{\zeta_t}{2} \mathrm{dist}(\mathbf{X}^t, \mathcal{C})^2 + \frac{\hat{L}_t \eta_t^2}{2} n D^2$$

$$\leq (1 - \eta_t) \left( \hat{F}_{t-1}(\mathbf{X}^t) - \hat{F}(\mathbf{X}^*) \right) + \frac{\hat{L}_t \eta_t^2}{2} n D^2 \quad (23)$$

where $\zeta_t := \lambda_t \left( \frac{\lambda_t}{\lambda_{t-1}} (1 - \eta_t) - 1 \right)$. In the last line, we used the fact that $\zeta_t < 0$ for $\eta_t = \frac{2}{t+1}$ and $\lambda_t = \lambda_0 \sqrt{t+1}$. By recursively applying this inequality, we get

$$\hat{F}_t(\mathbf{X}^{t+1}) - \hat{F}(\mathbf{X}^*) \leq 2n D^2 \left( \frac{L/n}{t+1} + \frac{\lambda_0}{\sqrt{t+1}} \right). \quad (24)$$

From the Lagrange saddle point theory, we know that the following bound holds $\forall \mathbf{X} \in \mathcal{D}$ and $\forall \mathbf{R} \in \mathcal{C}$

$$\hat{F}(\mathbf{X}^*) \leq \mathcal{L}(\mathbf{X}, \mathbf{R}, \mathbf{Y}^\star) = \hat{F}(\mathbf{X}) + \langle \mathbf{Y}^\star, \mathbf{X} - \mathbf{R} \rangle \leq \hat{F}(\mathbf{X}) + \|\mathbf{Y}^\star\| \|\mathbf{X} - \mathbf{R}\|, \quad (25)$$

since $\mathbf{X}^{t+1} \in \mathcal{D}$, we can write

$$\hat{F}(\mathbf{X}^{t+1}) - \hat{F}(\mathbf{X}^*) \geq - \min_{\mathbf{R} \in \mathcal{C}} \|\mathbf{Y}^\star\| \|\mathbf{X}^{t+1} - \mathbf{R}\| = -\|\mathbf{Y}^\star\| \mathrm{dist}(\mathbf{X}^{t+1}, \mathcal{C}). \quad (26)$$

Now, we can write

$$\hat{F}(\mathbf{X}^{t+1}) - \hat{F}(\mathbf{X}^*) \leq \underbrace{\hat{F}(\mathbf{X}^{t+1}) - \hat{F}(\mathbf{X}^*) + \frac{\lambda_t}{2} \mathrm{dist}(\mathbf{X}^{t+1}, \mathcal{C})^2}_{= \hat{F}_t(\mathbf{X}^{t+1}) - \hat{F}(\mathbf{X}^*)} \leq 2n D^2 \left( \frac{L/n}{t+1} + \frac{\lambda_0}{\sqrt{t+1}} \right).$$

$$(27)$$

where the last inequality comes from Equation (24). Combining this with Equation (26), we get

$$-\|\mathbf{Y}^\star\| \mathrm{dist}(\mathbf{X}^{t+1}, \mathcal{C}) + \frac{\lambda_t}{2} \mathrm{dist}(\mathbf{X}^{t+1}, \mathcal{C})^2 \leq 2n D^2 \left( \frac{L/n}{t+1} + \frac{\lambda_0}{\sqrt{t+1}} \right)$$

$$\leq \frac{2n D^2 \lambda_0 (L/n + \lambda_0)}{\lambda_t}. \quad (28)$$

Solving the quadratic inequality in terms of $\mathrm{dist}(\mathbf{X}^{t+1}, \mathcal{C})$, we have

$$\mathrm{dist}(\mathbf{X}^{t+1}, \mathcal{C}) \leq \frac{1}{\lambda_t} \left( \|\mathbf{Y}^\star\| + \sqrt{\|\mathbf{Y}^\star\|^2 + 4n D^2 \lambda_0 (L/n + \lambda_0)} \right)$$

$$\leq \frac{2}{\lambda_0 \sqrt{t+1}} \left( \|\mathbf{Y}^\star\| + D \sqrt{\lambda_0 (L + n \lambda_0)} \right). \quad (29)$$

Using convexity of $\hat{F}(.)$, we can write

$$\hat{F}(\bar{\mathbf{X}}^t) - \hat{F}(\mathbf{X}^*) \leq \hat{F}(\mathbf{X}^t) - \hat{F}(\mathbf{X}^*) + \langle \nabla \hat{F}(\bar{\mathbf{X}}^t), \bar{\mathbf{X}}^t - \mathbf{X}^t \rangle$$

$$\leq \hat{F}(\mathbf{X}^t) - \hat{F}^* + \|\nabla \hat{F}(\bar{\mathbf{X}}^t)\|_F \cdot \|\bar{\mathbf{X}}^t - \mathbf{X}^t\|_F$$

$$= \hat{F}(\mathbf{X}^t) - \hat{F}^* + \|\nabla \hat{F}(\bar{\mathbf{X}}^t)\|_F \cdot \mathrm{dist}(\mathbf{X}^t, \mathcal{C})$$

$$\leq \hat{F}(\mathbf{X}^t) - \hat{F}^* + \frac{G}{\sqrt{n}} \cdot \mathrm{dist}(\mathbf{X}^t, \mathcal{C})$$

$$\leq 2nD^2 \left( \frac{L/n}{t} + \frac{\lambda_0}{\sqrt{t}} \right) + \frac{G}{\sqrt{n}} \frac{2}{\lambda_0 \sqrt{t}} \left( \|\mathbf{Y}^\star\| + D\sqrt{\lambda_0(L + n\lambda_0)} \right), \quad (30)$$

where in the fourth line we used boundedness of the gradient with $G$ and in the last line we used bounds (27) and (29). Finally, we use the definition of $\hat{F}$ to derive our bound for $F(\bar{\mathbf{x}}^t) - F^*$

$$\hat{F}(\bar{\mathbf{X}}^t) - \hat{F}(\mathbf{X}^*) = \frac{1}{n} \sum_{i=1}^n f_i(\bar{\mathbf{X}}^t \mathbf{e}_i) - \frac{1}{n} \sum_{i=1}^n f_i(\mathbf{X}^* \mathbf{e}_i)$$

$$= \frac{1}{n} \sum_{i=1}^n f_i(\bar{\mathbf{x}}^t) - \frac{1}{n} \sum_{i=1}^n f_i(\mathbf{x}^*)$$

$$= F(\bar{\mathbf{x}}^t) - F^*$$

$$\leq \frac{2D^2 L}{t} + \frac{1}{\sqrt{t}} \left( 2nD^2\lambda_0 + \frac{2G}{\sqrt{n}\lambda_0} \left( \|\mathbf{Y}^\star\| + D\sqrt{\lambda_0(L + n\lambda_0)} \right) \right). \quad (31)$$

This completes the proof. $\qquad\square$

## C Convergence Analysis of Algorithm 1 (non-convex case)

To prove Theorem 2, we generalize Theorem 3.3 in [Yurtsever et al., 2018] for smooth and non-convex $F$.

### C.1 Proof of Theorem 2

The goal is to bound the Frank-Wolfe gap $h^t := \max_{\mathbf{U} \in \mathcal{D}^n} \langle \mathbf{U} - \mathbf{X}, -\nabla \hat{F}_t(\mathbf{X}) \rangle$ and show convergence of Algorithm 1 in the non-convex case to a feasible stationary point. Theorem 3 provides this results.

**Theorem 3.** *Consider problem* (1) *with L-smooth and non-convex loss functions* $f_i$. *Suppose that the sequence* $\{\bar{\mathbf{x}}^t\}$ *is generated by* FEDFW *with the fixed step-size* $\eta_t = T^{-2/3}$ *penalty parameter* $\lambda_t = \lambda_0 T^{1/3}$ *for some* $\lambda_0 > 0$. *Then,*

$$h^{\tilde{t}} \leq \mathcal{P}(T) = \mathcal{O}\left( \frac{1}{T^{\frac{1}{3}}} \right) \quad (32)$$

$$\mathrm{dist}(\mathbf{X}^{\tilde{t}}, \mathcal{C}) \leq \mathcal{Q}(T) = \mathcal{O}\left( \frac{1}{T^{\frac{1}{3}}} \right), \quad (33)$$

*where*

$$\mathcal{P}(T) := \left( \mathcal{E} + \frac{nD^2\lambda_0}{2} \right) \frac{1}{T^{\frac{1}{3}}} + \frac{LD^2}{2} \frac{1}{T^{\frac{2}{3}}}, \quad \mathcal{Q}(T) := \frac{G}{\lambda_0\sqrt{n}} \frac{1}{T^{\frac{1}{3}}} + \sqrt{\left( \mathcal{E} + \frac{D^2\lambda_0}{2} \right) \frac{1}{T^{\frac{2}{3}}} + \frac{L_f D^2}{2} \frac{1}{T}}, \quad (34)$$

*and* $\mathcal{E}$ *is the initial error.*

*Proof.* In what follows, we prove Equation (32) in part **(a)** and Equation (33) in part **(b)**.

**(a)** Using the smoothness of $\hat{F}_t(.)$, Remark 5, and defining $\hat{L}_t := \frac{L}{n} + \lambda_t$, we have

$$\hat{F}_t(\mathbf{X}^{t+1}) \leq \hat{F}_t(\mathbf{X}^t) + \langle \nabla \hat{F}_t(\mathbf{X}^t), \mathbf{X}^{t+1} - \mathbf{X}^t \rangle + \frac{\hat{L}_t}{2} \|\mathbf{X}^{t+1} - \mathbf{X}^t\|_F^2$$

$$= \hat{F}_t(\mathbf{X}^t) + \eta_t \langle \nabla \hat{F}_t(\mathbf{X}^t), \mathbf{S}^t - \mathbf{X}^t \rangle + \frac{\hat{L}_t \eta_t^2}{2} \|\mathbf{S}^t - \mathbf{X}^t\|_F^2$$

$$\leq \hat{F}_t(\mathbf{X}^t) + \eta_t \langle \nabla \hat{F}_t(\mathbf{X}^t), \mathbf{S}^t - \mathbf{X}^t \rangle + \frac{\hat{L}_t \eta_t^2}{2} nD^2$$

$$= \hat{F}_t(\mathbf{X}^t) - \eta_t h^t + \frac{\hat{L}_t \eta_t^2}{2} nD^2. \tag{35}$$

where in the secomd line we used Equations (8) and (16), and the updating rule $\mathbf{x}_i^{t+1} = (1-\eta_t)\mathbf{x}_i^t + \eta_t \mathbf{s}_i^t$. In the third line we used boundedness of the $\mathcal{D}$, $\|\mathbf{y} - \mathbf{x}\| \leq D \; \forall \, \mathbf{x}, \mathbf{y} \in \mathcal{D}$. Now, considering assumptions for the non-convex case, $\eta_t = \eta = T^{-2/3}, \lambda_t = \lambda = \lambda_0 T^{1/3}, \hat{L}_t = \hat{L}$ and rearranging (35), taking the sum of both sides from 1 to $T$ and deviding by $\eta T$, we have

$$\frac{1}{T} \sum_{t=1}^T h^t \leq \frac{1}{\eta T} \sum_{t=1}^T \left( \hat{F}_t(\mathbf{X}^t) - \hat{F}_t(\mathbf{X}^{t+1}) \right) + \frac{\hat{L}\eta}{2} nD^2$$

$$= \frac{1}{\eta T} \sum_{t=1}^T \left( \hat{F}_t(\mathbf{X}^1) - \hat{F}_t(\mathbf{X}^{T+1}) \right) + \frac{\hat{L}\eta}{2} nD^2 \qquad (\text{telescoping series}). \tag{36}$$

Note that

$$\hat{F}_t(\mathbf{X}^1) - \hat{F}_t(\mathbf{X}^{T+1}) = \frac{1}{n} \sum_{i=1}^n f_i(\mathbf{X}^1 \mathbf{e}_i) + \frac{\lambda}{2} \text{dist}(\mathbf{X}^1, \mathcal{C})^2 - \frac{1}{n} \sum_{i=1}^n f_i(\mathbf{X}^{T+1} \mathbf{e}_i) - \frac{\lambda}{2} \text{dist}(\mathbf{X}^{T+1}, \mathcal{C})^2$$

$$\leq \frac{1}{n} \sum_{i=1}^n f_i(\mathbf{X}^1 \mathbf{e}_i) + \frac{\lambda}{2} \text{dist}(\mathbf{X}^1, \mathcal{C})^2 - \frac{1}{n} \sum_{i=1}^n f_i(\mathbf{X}^{T+1} \mathbf{e}_i)$$

$$\leq \frac{1}{n} \sum_{i=1}^n \left( f_i(\mathbf{X}^1 \mathbf{e}_i) - f_i(\mathbf{X}^{T+1} \mathbf{e}_i) \right) + \frac{\lambda}{2} \text{dist}(\mathbf{X}^1, \mathcal{C})^2$$

$$\leq \frac{1}{n} \sum_{i=1}^n \left( f_i(\mathbf{X}^1 \mathbf{e}_i) - f_i(\mathbf{X}^* \mathbf{e}_i) \right)$$

$$=: \mathcal{E}. \tag{37}$$

where in the forth line we used the assumption that initial conditions belong to $\mathcal{C}$ or $\mathbf{X}^1 \in \mathcal{C}$, therefore we have $\text{dist}(\mathbf{X}^1, \mathcal{C}) = 0$. Substituting Equation (37) into Equation (36) and using $\eta = T^{-\frac{2}{3}}, \lambda = \lambda_0 T^{\frac{1}{3}}, \hat{L} := \frac{L}{n} + \lambda$, we get

$$\frac{1}{T} \sum_{t=1}^T h^t \leq \frac{\mathcal{E}}{T \cdot T^{-\frac{2}{3}}} + \frac{(L/n + \lambda_0 T^{\frac{1}{3}}) T^{-\frac{2}{3}}}{2} nD^2$$

$$= \left( \mathcal{E} + \frac{nD^2 \lambda_0}{2} \right) \frac{1}{T^{\frac{1}{3}}} + \frac{LD^2}{2} \frac{1}{T^{\frac{2}{3}}}. \tag{38}$$

Now let us define $\tilde{t} \in \text{argmin}\{h^t\} \in [T]$, we can use Equation (38) to write

$$h^{\tilde{t}} \leq \left( \mathcal{E} + \frac{nD^2 \lambda_0}{2} \right) \frac{1}{T^{\frac{1}{3}}} + \frac{LD^2}{2} \frac{1}{T^{\frac{2}{3}}} =: \mathcal{P}(T), \tag{39}$$

which can be translated to $h^{\tilde{t}} \leq \mathcal{P}(T) = \mathcal{O}\left( \frac{1}{T^{\frac{1}{3}}} \right)$. This proves the first bound in Theorem 3.

**(b)** To prove that $\mathbf{X}^t$ converges to $\mathcal{C}$, we start with the definition of $h^{\tilde{t}}$

$$
\begin{aligned}
h^{\tilde{t}} &= \max_{\mathbf{U} \in \mathcal{D}^n} \langle \nabla \hat{F}_{\tilde{t}}(\mathbf{X}^{\tilde{t}}), \mathbf{X}^{\tilde{t}} - \mathbf{U} \rangle \\
&\geq \langle \nabla \hat{F}_{\tilde{t}}(\mathbf{X}^{\tilde{t}}), \mathbf{X}^{\tilde{t}} - \bar{\mathbf{X}}^{\tilde{t}} \rangle \\
&= \langle \frac{1}{n} \sum_{i=1}^{n} \nabla f_i(\mathbf{x}_i^{\tilde{t}}) \cdot \mathbf{e}_i^{\top} + \lambda(\mathbf{X}^{\tilde{t}} - \bar{\mathbf{X}}^{\tilde{t}}), \mathbf{X}^{\tilde{t}} - \bar{\mathbf{X}}^{\tilde{t}} \rangle \\
&= \langle \frac{1}{n} \sum_{i=1}^{n} \nabla f_i(\mathbf{x}_i^{\tilde{t}}) \cdot \mathbf{e}_i^{\top}, \mathbf{X}^{\tilde{t}} - \bar{\mathbf{X}}^{\tilde{t}} \rangle + \lambda \|\mathbf{X}^{\tilde{t}} - \bar{\mathbf{X}}^{\tilde{t}}\|_F^2 \\
&\geq -\|\frac{1}{n} \sum_{i=1}^{n} \nabla f_i(\mathbf{x}_i^{\tilde{t}}) \cdot \mathbf{e}_i^{\top}\|_F \cdot \|\mathbf{X}^{\tilde{t}} - \bar{\mathbf{X}}^{\tilde{t}}\|_F + \lambda \|\mathbf{X}^{\tilde{t}} - \bar{\mathbf{X}}^{\tilde{t}}\|_F^2 \quad \text{(Cauchy–Schwarz inequality)} \\
&= -\frac{G}{\sqrt{n}} \|\mathbf{X}^{\tilde{t}} - \bar{\mathbf{X}}^{\tilde{t}}\|_F + \lambda \|\mathbf{X}^{\tilde{t}} - \bar{\mathbf{X}}^{\tilde{t}}\|_F^2 \quad (\|\nabla f_i(\mathbf{x})\| \leq G). \tag{40}
\end{aligned}
$$

Using Equation (39) and Equation (40) , we can write

$$
-\frac{G}{\sqrt{n}} \|\mathbf{X}^{\tilde{t}} - \bar{\mathbf{X}}^{\tilde{t}}\|_F + \lambda \|\mathbf{X}^{\tilde{t}} - \bar{\mathbf{X}}^{\tilde{t}}\|_F^2 \leq \mathcal{P}(T), \tag{41}
$$

and solve the quadratic inequality (41)

$$
\begin{aligned}
\|\mathbf{X}^{\tilde{t}} - \bar{\mathbf{X}}^{\tilde{t}}\|_F &\leq \frac{G/\sqrt{n} + \sqrt{(G/\sqrt{n})^2 + 4\lambda \mathcal{P}(T)}}{2\lambda} \\
&= \frac{G/\sqrt{n} + \sqrt{\left(G/\sqrt{n} + 2\sqrt{\lambda \mathcal{P}(T)}\right)^2 - 2G/\sqrt{n}\sqrt{\lambda \mathcal{P}(T)}}}{2\lambda} \\
&\leq \frac{G/\sqrt{n} + \sqrt{\lambda \mathcal{P}(T)}}{\lambda}. \tag{42}
\end{aligned}
$$

Plugging $\lambda = \lambda_0 T^{\frac{1}{3}}$ and $\mathcal{P}(T)$, we get

$$
\|\mathbf{X}^{\tilde{t}} - \bar{\mathbf{X}}^{\tilde{t}}\|_F \leq \frac{G}{\lambda_0 \sqrt{n}} \frac{1}{T^{\frac{1}{3}}} + \sqrt{\left(\mathcal{E} + \frac{D^2 \lambda_0}{2}\right) \frac{1}{T^{\frac{2}{3}}} + \frac{L_f D^2}{2} \frac{1}{T}} =: \mathcal{Q}(T). \tag{43}
$$

Note that $\|\mathbf{X}^{\tilde{t}} - \bar{\mathbf{X}}^{\tilde{t}}\|_F$ can be interpreted as the distance of $\mathbf{X}^{\tilde{t}}$ to $\mathcal{C}$

$$
\text{dist}(\mathbf{X}^{\tilde{t}}, \mathcal{C}) \leq \mathcal{Q}(T) = \mathcal{O}\left(\frac{1}{T^{\frac{1}{3}}}\right). \tag{44}
$$

This completes the proof of Theorem 3. $\qquad\square$

Next, we use Theorem 3 to show the result in Theorem 2. Starting from the definition of $h^{\tilde{t}}$, we write

$$
\begin{aligned}
h^{\tilde{t}} &= \max_{\mathbf{U} \in \mathcal{D}^n} \langle \nabla \hat{F}_{\tilde{t}}(\mathbf{X}^{\tilde{t}}), \mathbf{X}^{\tilde{t}} - \mathbf{U} \rangle \\
&= \max_{\mathbf{U} \in \mathcal{D}^n} \left\{ \langle \nabla \bar{F}(\mathbf{X}^{\tilde{t}}), \mathbf{X}^{\tilde{t}} - \mathbf{U} \rangle + \lambda \langle \mathbf{X}^{\tilde{t}} - \bar{\mathbf{X}}^{\tilde{t}}, \mathbf{X}^{\tilde{t}} - \mathbf{U} \rangle \right\} \\
&= \max_{\mathbf{U} \in \mathcal{D}^n} \left\{ \langle \nabla \bar{F}(\mathbf{X}^{\tilde{t}}), \mathbf{X}^{\tilde{t}} - \mathbf{U} \rangle + \lambda \langle \mathbf{X}^{\tilde{t}} - \bar{\mathbf{X}}^{\tilde{t}}, \bar{\mathbf{X}}^{\tilde{t}} - \mathbf{U} \rangle \right\} + \underbrace{\lambda \langle \mathbf{X}^{\tilde{t}} - \bar{\mathbf{X}}^{\tilde{t}}, \mathbf{X}^{\tilde{t}} - \bar{\mathbf{X}}^{\tilde{t}} \rangle}_{=\|\mathbf{X}^{\tilde{t}} - \bar{\mathbf{X}}^{\tilde{t}}\|_F^2 \geq 0} \\
&\geq \max_{\mathbf{U} \in \mathcal{D}^n \cap \mathcal{C}} \left\{ \langle \nabla \bar{F}(\mathbf{X}^{\tilde{t}}), \mathbf{X}^{\tilde{t}} - \mathbf{U} \rangle + \underbrace{\lambda \langle \mathbf{X}^{\tilde{t}} - \bar{\mathbf{X}}^{\tilde{t}}, \bar{\mathbf{X}}^{\tilde{t}} - \mathbf{U} \rangle}_{\geq 0} \right\} \quad \text{(projection is non-expansive)} \\
&\geq \max_{\mathbf{U} \in \mathcal{D}^n \cap \mathcal{C}} \left\{ \langle \nabla \bar{F}(\mathbf{X}^{\tilde{t}}), \mathbf{X}^{\tilde{t}} - \mathbf{U} \rangle \right\} \\
&\geq \max_{\mathbf{U} \in \mathcal{D}^n \cap \mathcal{C}} \Big\{ \underbrace{\langle \nabla \bar{F}(\mathbf{X}^{\tilde{t}}) - \nabla \bar{F}(\bar{\mathbf{X}}^{\tilde{t}}), \mathbf{X}^{\tilde{t}} - \mathbf{U} \rangle}_{\geq -L\|\mathbf{X}^{\tilde{t}} - \bar{\mathbf{X}}^{\tilde{t}}\|\|\mathbf{X}^{\tilde{t}} - \mathbf{U}\| \geq -LD\|\mathbf{X}^{\tilde{t}} - \bar{\mathbf{X}}^{\tilde{t}}\|} + \langle \nabla \bar{F}(\bar{\mathbf{X}}^{\tilde{t}}), \bar{\mathbf{X}}^{\tilde{t}} - \mathbf{U} \rangle \Big\} + \underbrace{\langle \nabla \bar{F}(\bar{\mathbf{X}}^{\tilde{t}}), \mathbf{X}^{\tilde{t}} - \bar{\mathbf{X}}^{\tilde{t}} \rangle}_{\geq -G \cdot \|\mathbf{X}^{\tilde{t}} - \bar{\mathbf{X}}^{\tilde{t}}\|_F} \\
&\geq \max_{\mathbf{U} \in \mathcal{D}^n \cap \mathcal{C}} \left\{ \langle \nabla \bar{F}(\bar{\mathbf{X}}^{\tilde{t}}), \bar{\mathbf{X}}^{\tilde{t}} - \mathbf{U} \rangle \right\} - (LD + G) \|\mathbf{X}^{\tilde{t}} - \bar{\mathbf{X}}^{\tilde{t}}\|_F \\
&= \max_{\mathbf{U} \in \mathcal{D}^n \cap \mathcal{C}} \left\{ \langle \nabla \bar{F}(\bar{\mathbf{X}}^{\tilde{t}}), \bar{\mathbf{X}}^{\tilde{t}} - \mathbf{U} \rangle \right\} - (LD + G) \cdot \mathrm{dist}(\mathbf{X}^{\tilde{t}}, \mathcal{C}). \qquad (45)
\end{aligned}
$$

Using the upper bounds on $h^{\tilde{t}}$, $\mathrm{dist}(\mathbf{X}^{\tilde{t}}, \mathcal{C})$ and rearranging the last equation above, we have

$$
\max_{\mathbf{U} \in \mathcal{D}^n \cap \mathcal{C}} \left\{ \langle \nabla \bar{F}(\bar{\mathbf{X}}^{\tilde{t}}), \bar{\mathbf{X}}^{\tilde{t}} - \mathbf{U} \rangle \right\} \leq (LD + G) \mathcal{Q}(T) + \mathcal{P}(T) = \mathcal{O}\left( \frac{1}{T^{\frac{1}{3}}} \right). \qquad (46)
$$

Last equation can be re-writen as

$$
\min_{t \in [T]} \max_{\mathbf{u} \in \mathcal{D} \cap \mathcal{C}} \left\{ \langle \nabla F(\bar{\mathbf{x}}^t), \bar{\mathbf{x}}^t - \mathbf{u} \rangle \right\} \leq \mathcal{O}\left( \frac{1}{T^{\frac{1}{3}}} \right). \qquad (47)
$$

This concludes the proof. $\qquad\qquad\square$

## D  FedFW+

This section introduces an extension of FEDFW with augmented Lagrangian dual steps. We call this new variant FEDFW+ and present the details in Algorithm 2. The new steps in FEDFW+ compared against FEDFW is highlighted with the red font in Algorithm 2.

---

**Algorithm 2** FEDFW+: Federated Frank-Wolfe Algorithm via augmented Lagrangian

---

    **set** $\mathbf{x}_i^1 \in \mathbb{R}^p$, $\forall i \in [n]$, $\lambda_t$, $\eta_t$, $\bar{\mathbf{x}}^1 = \frac{1}{n} \sum_{i=1}^n \mathbf{x}_i^1$
    **for** round $t = 1, 2, \ldots, T$ **do**

        — **Client**-level local training —————————————
        **for** client $i = 1, 2, \ldots, n$ **do**
            $\color{red}{\mathbf{y}_i^{t+1} = \mathbf{y}_i^t + \lambda_0 (\mathbf{x}_i^t - \bar{\mathbf{x}}^t)}$
            $\mathbf{g}_i^t = \frac{1}{n} \nabla f_i(\mathbf{x}_i^t) + \lambda_t (\mathbf{x}_i^t - \bar{\mathbf{x}}^t) + \color{red}{\mathbf{y}_i^{t+1}}$
            $\mathbf{s}_i^t = \arg\min \{ \langle \mathbf{g}_i^t, \mathbf{x} \rangle : \mathbf{x} \in \mathcal{D} \}$
            $\mathbf{x}_i^{t+1} = (1 - \eta_t) \mathbf{x}_i^t + \eta_t \mathbf{s}_i^t$
            Client communicates $\mathbf{s}_i^t$ to the server.
        **end for**

        — **Server**-level aggregation —————————————
        $\bar{\mathbf{x}}^{t+1} = (1 - \eta_t) \bar{\mathbf{x}}^t + \eta_t \left( \frac{1}{n} \sum_{i=1}^n \mathbf{s}_i^t \right)$
        Server communicates $\bar{\mathbf{x}}^{t+1}$ to the clients.

    **end for**

---

