# OpenReview forum: "Federated Frank-Wolfe Algorithm"
_NeurIPS.cc/2022/Workshop/Federated_Learning — FL-NeurIPS 2022 Poster_

### Official Review · Reviewer_rG4L · 2022-10-18
**Federated Frank-Wolfe Algorithms**

This paper develops a federated version of Frank-Wolfe method and analyzes its convergence.

The paper is well-written and has solid results. The limitation is that it is not well-motivated since not many problems in FL has such constraints.

---

### Official Review · Reviewer_Tvww · 2022-10-19
**Good theoretical paper with poor experiments.**

This paper introduces the Federated Frank-Wolfe algorithm for solving constrained optimization problem in distributed regime. Authors provided good explanations why previous attempts of applying Frank-Wolfe algorithm were not successful. The theory seems to be sound. However, this paper compares only two versions of proposed methods and it lacks comparison with other baselines and advanced algorithms.

Overall, I think this paper is good to be presented.

---

### Decision · Program_Chairs · 2022-10-20

Accept (Poster)